# ASSESSING GENERALIZATION IN
# DEEP REINFORCEMENT LEARNING

## ABSTRACT

Deep reinforcement learning (RL) has achieved breakthrough results on many tasks, but has been shown to be sensitive to system changes at test time. As a result, building deep RL agents that generalize has become an active research area. Our aim is to catalyze and streamline community-wide progress on this problem by providing the first benchmark and a common experimental protocol for investigating generalization in RL. Our benchmark contains a diverse set of environments and our evaluation methodology covers both in-distribution and out-of-distribution generalization. To provide a set of baselines for future research, we conduct a systematic evaluation of deep RL algorithms, including those that specifically tackle the problem of generalization. We find that classic deep RL algorithms often perform as well or better than those that aim to generalize.

## 1 INTRODUCTION

Deep reinforcement learning (RL) has emerged as an important family of techniques that may support the development of intelligent systems that learn to accomplish goals in a variety of complex real-world environments (Mnih et al., 2015; Arulkumaran et al., 2017). A desirable characteristic of such intelligent systems is the ability to function in diverse environments, including ones that have never been encountered before. Yet, deep RL algorithms are commonly trained and evaluated on a fixed environment. The algorithms are evaluated in terms of their ability to optimize a policy in a complex environment, rather than their ability to learn a representation that generalizes to previously unseen circumstances. Indeed, their sensitivity to even subtle changes in the environment and the dangers of overfitting to a specific environment have been noted in the literature (Rajeswaran et al., 2017b; Henderson et al., 2018; Zhang et al., 2018; Whiteson et al., 2011).

Generalization is often regarded as an essential characteristic of advanced intelligent systems and a central issue in AI research (Lake et al., 2017; Marcus, 2018; Dietterich, 2017). It refers to both interpolation to environments similar to those seen during training and extrapolation outside the training data distribution. The latter is particularly challenging but is crucial to the deployment of systems in the real world.

Generalization in deep RL has been recognized as an important problem and is under active investigation (Rajeswaran et al., 2017a; Pinto et al., 2017; Kansky et al., 2017; Yu et al., 2017; Wang et al., 2016; Duan et al., 2016b; Sung et al., 2017; Clavera et al., 2018; Sæmundsson et al., 2018). However, each work uses a different set of environments and experimental protocols. For example, Kansky et al. (2017) propose a graphical model architecture, evaluating on variations of the Atari game Breakout. Rajeswaran et al. (2017a) propose training on a distribution of domains in risk-averse manner and evaluate on two continuous control tasks from MuJoCo (Hopper and HalfCheetah). Duan et al. (2016b) aim to learn a policy that automatically adapts to the environment dynamics and evaluate on bandits, tabular Markov decision processes, and maze navigation. Sæmundsson et al. (2018) combine learning a hierarchical latent model for the environment dynamics and model predictive control, evaluating on two continuous control tasks (cart-pole swing-up and double-pendulum swing-up).

What appears to be missing is a common testbed for evaluating generalization in deep RL: a clearly defined set of tasks, metrics, and baselines that can support concerted community-wide progress. In other words, research on generalization in deep RL has not yet adopted the 'common task framework', a proven catalyst of progress (Donoho, 2015). Only by creating such testbeds and evaluating

on them can we fairly compare and contrast the merits of different algorithms and accurately measure progress made on the problem.

Our contribution is to establish a reproducible framework for investigating generalization in deep RL, with the hope that it will catalyze progress on this problem, and to present an empirical evaluation of generalization in deep RL algorithms as a baseline. We select a diverse but manageable set of environments, comprising classic control problems and MuJoCo locomotion tasks, built on top of OpenAI Gym for ease of adoption. Like Rajeswaran et al. (2017a) and others, we focus on generalization to changes in the system dynamics, which is implemented by specifying degrees of freedom (parameters) along which the environment specifications can be varied. Significantly, we test generalization in two regimes: interpolation and extrapolation. Interpolation implies that agents should perform well in test environments where parameters are similar to those seen during training. Extrapolation requires agents to perform well in test environments where parameters are different from those seen during training.

To provide the community with a set of clear baselines, we evaluate two deep RL algorithms on all environments and under different combinations of training and testing regimes. We chose one algorithm from each of the two major families: A2C from the actor-critic family and PPO from the policy gradient family. Using the same experimental protocol, we also evaluate two schemes for tackling generalization in deep RL: EPOpt, which learns a policy that is robust to environment changes by maximizing expected reward over the most difficult of a distribution of environment parameters, and $RL^2$, which learns a policy that can adapt to the environment at hand by taking into account the trajectory it sees. Because each scheme is constructed based on existing deep RL algorithms, our evaluation is of four algorithms: EPOpt-A2C, EPOpt-PPO, $RL^2$-A2C, and $RL^2$-PPO. We analyze the results and draw conclusions that can guide future work on generalization in deep RL. The experimental results confirm that extrapolation is more difficult than interpolation and show that the 'vanilla' deep RL algorithms (A2C and PPO) were able to interpolate fairly successfully. Somewhat surprisingly, they interpolate and extrapolate better than their EPOpt and $RL^2$ variants, with the exception of EPOpt-PPO. $RL^2$-A2C and $RL^2$-PPO proved to be difficult to train and were unable to reach the level of performance of the other algorithms given the same amount of training resources.

## 2 RELATED WORK

**Generalization in RL**. There are two main approaches to generalization in RL: learning policies that are robust to environment variations, or learning policies that adapt to such variations. A popular approach to learn a robust policy is to maximize a risk-sensitive objective, such as the conditional value at risk (Tamar et al., 2015), over a distribution of environments. Morimoto & Doya (2001) maximize the minimum reward over possible disturbances, proposing robust versions of the actor-critic and value gradient methods in a control theory framework. This maximin objective is utilized by others in the context where environment changes are modeled by uncertainties in the transition probability distribution function of a Markov decision process. Nilim & Ghaoui (2004) assume that the set of possible transition probability distribution functions are known, while Lim et al. (2013) and Roy et al. (2017) estimate it using sampled trajectories from the distribution of environments of interest. A recent representative of this approach applied to deep RL is the EPOpt algorithm (Rajeswaran et al., 2017a), which maximizes the conditional value at risk, i.e. expected reward over the subset of environments with lowest expected reward. EPOpt has the advantage that it can be used in conjunction with any RL algorithm. Adversarial training has also been proposed to learn a robust policy; for MuJoCo locomotion tasks, Pinto et al. (2017) trains an adversary that tries to destabilize the agent during training.

A robust policy may sacrifice performance on many environment variants in order to not fail on a few. Thus, an alternative, recently popular approach to generalization in RL is to learn a policy that can adapt to the environment at hand (Yu et al., 2017). To do so, a number of algorithms learn an embedding for each environment variant using trajectories sampled from that environment, which is input into a policy. Then, at test time, the current trajectory can be used to compute an embedding for the current environment, enabling automatic adaptation of the policy. Duan et al. (2016b), Wang et al. (2016), Sung et al. (2017), and Mishra et al. (2018), which differ mainly in the way embeddings are computed, consider model-free RL by letting the embedding be input into a policy and/or value

function. Clavera et al. (2018) consider model-based RL, in which the embedding is input into a dynamics model and actions are selected using model predictive control. Under a similar setup, Sæmundsson et al. (2018) utilize probabilistic dynamics models and inference.

This literature review has focused on RL algorithms for generalization that do not require updating the learned model or policy at test time, in keeping with our benchmark's evaluation procedure. There has been work on generalization in RL that utilize such updates, primarily under the umbrellas of transfer learning, multi-task learning, and meta-learning. Taylor & Stone (2009) surveys transfer learning in RL where a fixed test environment is considered, with Rusu et al. (2016) being an example of recent work on that problem using deep networks. Ruder (2017) provides a survey of multi-task learning in general, which, different from our problem of interest, considers a fixed finite population of tasks. Finn et al. (2017) present a meta-learning formulation of generalization in RL and Al-Shedivat et al. (2018) extend it for continuous adaptation in non-stationary environments.

**Empirical methodology in deep RL**. Shared open-source software infrastructure, which enables reproducible experiments, has been crucial to the success of deep RL. The deep RL research community uses simulation frameworks, including OpenAI Gym (Brockman et al., 2016), the Arcade Learning Environment (Bellemare et al., 2013; Machado et al., 2017), DeepMind Lab (Beattie et al., 2016), and VizDoom (Kempka et al., 2016). The MuJoCo physics simulator (Todorov et al., 2012) has been influential in standardizing a number of continuous control tasks. For ease of adoption, our work builds on OpenAI Gym and MuJoCo tasks, allowing variations in the environment specifications in order to study generalization. OpenAI recently released a benchmark for transfer learning in RL (Nichol et al., 2018), in which the goal is to train an agent to play new levels of a video game with fine-tuning at test time. In contrast, our benchmark does not allow fine-tuning and focuses on control tasks.

Our work also follows in the footsteps of a number of empirical studies of reinforcement learning algorithms, which have primarily focused on the case where the agent is trained and tested on a fixed environment. Henderson et al. (2018) investigate reproducibility in deep RL, testing state-of-the-art algorithms on four MuJoCo tasks: HalfCheetah, Hopper, Walker2d, and Swimmer. They show that results may be quite sensitive to hyperparameter settings, initialization, random seeds, and other implementation details, indicating that care must be taken not to overfit to a particular environment. The problem of overfitting in RL was recognized earlier by Whiteson et al. (2011), who propose an evaluation methodology based on training and testing on multiple environments sampled from a distribution and experiment with three classic environments: MountainCar, Acrobot, and puddle world. Nair et al. (2015) evaluate generalization with respect to different starting points in Atari games. Duan et al. (2016a) present a benchmark suite of continuous control tasks and conduct a systematic evaluation of reinforcement learning algorithms on those tasks. They consider generalization in terms of interpolation on a subset of their tasks. In contrast to these works, we address a greater variety of tasks, extrapolation as well as interpolation, and algorithms for learning deep RL agents that generalize.

## 3 NOTATION

In RL, environments are formulated in terms of Markov Decision Processes (MDPs) (Sutton & Barto, 2017). An MDP $M$ is defined by the tuple $(\mathbb{S}, \mathbb{A}, p, r, \gamma, \rho_0, T)$ where $\mathbb{S}$ is the set of possible states, $\mathbb{A}$ is the set of actions, $p : \mathbb{S} \times \mathbb{A} \times \mathbb{S} \to \mathbb{R}_{\geq 0}$ is the transition probability distribution function, $r : \mathbb{S} \times \mathbb{A} \to \mathbb{R}$ is the reward function, $\gamma$ is the discount factor, $\rho_0 : \mathbb{S} \to \mathbb{R}_{\geq 0}$ is the initial state distribution at the beginning of each episode, and $T$ is the time horizon per episode. Generalization to environment variations is usually characterized as generalization to changes in $p$ and $r$; our benchmark considers changes in $p$.

Let $s_t$ and $a_t$ be the state and action taken at time $t$. At the beginning of each episode, $s_0 \sim \rho_0(s_0)$. Under a policy $\pi$ stochastically mapping a sequence of states to actions, $a_t \sim \pi(a_t \mid s_t, \cdots, s_0)$ and $s_{t+1} \sim p(s_{t+1} \mid a_t)$, giving a trajectory $\{s_t, a_t, r(s_t, a_t)\}$, $t = 0, 1, \cdots$. RL algorithms, taking the MDP as fixed, learn $\pi$ to maximize the expected reward over an episode $J_M(\pi) = \mathbb{E}^\pi \left[ \sum_{t=0}^{T} \gamma^t r_t \right]$, where $r_t = r(s_t, a_t)$. They often utilize the concepts

of a value function $v_M^\pi(\boldsymbol{s}) = \mathbb{E}^\pi \left[ \sum_{t=0}^T \gamma^t r(\boldsymbol{s}_t, \boldsymbol{a}_t) \mid \boldsymbol{s}_0 = \boldsymbol{s} \right]$ and a state-action value function $Q_M^\pi(\boldsymbol{s}, \boldsymbol{a}) = \mathbb{E}^\pi \left[ \sum_{t=0}^T \gamma^t r(\boldsymbol{s}_t, \boldsymbol{a}_t) \mid \boldsymbol{s}_0 = \boldsymbol{s}, \boldsymbol{a}_0 = \boldsymbol{a} \right]$.

# 4 ALGORITHMS

We first evaluate 'vanilla' deep RL algorithms from two main categories: actor-critic and policy gradient. From the actor-critic family, we chose A2C (Mnih et al., 2016), and from the policy gradient family we chose PPO (Schulman et al., 2017).[1] These algorithms are oblivious to variations in the environment; they were not designed with generalization in mind. We also include recently-proposed algorithms that are designed to be able to generalize: EPOpt (Rajeswaran et al., 2017a) from the robust approaches and RL$^2$ (Duan et al., 2016b) from the adaptive approaches. Both these methods are built on top of 'vanilla' deep RL algorithms, so for completeness we evaluate a Cartesian product of the algorithms for generalization and the 'vanilla' algorithms: EPOpt-A2C, EPOpt-PPO, RL$^2$-A2C, and RL$^2$-PPO. Next we briefly summarize A2C, PPO, EPOpt, and RL$^2$, using the notation in Section 3.

**Advantage Actor-Critic (A2C)**. A2C involves the interplay of two optimizers; a critic learns a parametric value function, while an actor utilizes that value function to learn a parametric policy that maximizes expected reward. At each iteration, trajectories are generated using the current policy, with the environment and hidden states of the value function and policy reset at the end of each episode. Then, the policy and value function parameters are updated using RMSProp (Hinton et al., 2012), with an entropy term added to the policy objective function in order to encourage exploration. We use an implementation from OpenAI Baselines (Dhariwal et al., 2017).

**Proximal Policy Optimization (PPO)**. PPO aims to learn a sequence of monotonically improving parametric policies by maximizing a surrogate for the expected reward via gradient ascent, cautiously bounding the improvement achieved at each iteration. At iteration $i$, trajectories are generated using the current policy $\pi_{\theta_i}$, with the environment and hidden states of the policy reset at the end of each episode. The following objective is then maximized with respect to $\theta$ using Adam (Kingma & Ba, 2015):

$$\mathbb{E}_{\boldsymbol{s} \sim \rho_{\theta_i}, \boldsymbol{a} \sim \pi_{\theta_i}} \min \left[ \ell_\theta(\boldsymbol{a}, \boldsymbol{s}) A_{\pi_{\theta_i}}(\boldsymbol{s}, \boldsymbol{a}), m_\theta(\boldsymbol{a}, \boldsymbol{s}) A_{\pi_{\theta_i}}(\boldsymbol{s}, \boldsymbol{a}) \right]$$

where $\rho_{\theta_i}$ are the expected visitation frequencies under $\pi_{\theta_i}$, $\ell_\theta(\boldsymbol{a}, \boldsymbol{s}) = \pi_\theta(\boldsymbol{a} \mid \boldsymbol{s}) / \pi_{\theta_i}(\boldsymbol{a} \mid \boldsymbol{s})$, $m_\theta$ equals $\ell_\theta(\boldsymbol{a}, \boldsymbol{s})$ clipped to the interval $[1 - \delta, 1 + \delta]$ with $\delta \in (0, 1)$, and $A_{\pi_{\theta_i}}(\boldsymbol{s}, \boldsymbol{a}) = Q_M^{\pi_{\theta_i}}(\boldsymbol{s}, \boldsymbol{a}) - v_M^{\pi_{\theta_i}}(\boldsymbol{s})$. Again, we use an implementation from OpenAI Baselines, PPO2.

**Ensemble Policy Optimization (EPOpt)**. To generalize over a distribution of environments (MDPs) $p(M)$, we would like to learn a policy that maximizes the expected reward over the distribution, $\mathbb{E}_{M \sim p(M)}^\pi [J_M(\pi)]$. In order to obtain a policy that is also robust to out-of-distribution environments, EPOpt instead maximizes the expected reward over the $\epsilon \in (0, 1]$ fraction of environments with worst expected reward:

$$\mathbb{E}_{M \sim p(M)}^\pi [J_M(\pi) \leq y] \quad \text{where} \quad P_{M \sim p(M)}(J_M(\pi) \leq y) = \epsilon.$$

At each iteration, the algorithm generates a number of complete episodes according to the current policy where at the end of each episode a new environment is sampled from $p(M)$ and reset. (As in A2C and PPO, at the end of each episode the hidden states of the policy and value function are reset.) It keeps the $\epsilon$ fraction of episodes with lowest reward and uses them to update the policy with some RL algorithm (TRPO (Schulman et al., 2015) in the paper). We instead use A2C and PPO, building our implementation of EPOpt on top of the implementations of A2C and PPO.

**RL$^2$**. To learn a policy that can adapt to the dynamics of the environment at hand, RL$^2$ models the policy and value functions as a recurrent neural network (RNN) with the current trajectory as input, not just the sequence of states. The hidden states of the RNN may be viewed as an environment embedding. Specifically, for the RNN the inputs at time $t$ are $\boldsymbol{s}_t$, $\boldsymbol{a}_{t-1}$, $r_{t-1}$, and $d_{t-1}$, where $d_{t-1}$

---

[1] We carried out preliminary experiments on other deep RL algorithms including A3C, TRPO, and ACKTR. A2C and A3C/ACKTR had similar qualitative results, as did PPO and TRPO.

is a Boolean variable indicating whether the episode ended after taking action $a_{t-1}$; the output is $a_t$ and the hidden states are updated to $h_{t+1}$. Like the other algorithms, at each iteration trajectories are generated using the current policy with the environment state reset at the end of each episode. However, unlike the other algorithms, a new environment is sampled from $p(M)$ only at the end of every $N$ episodes, which we call a trial. ($N = 2$ in our experiments.) Likewise, the hidden states of the policy and value functions are reinitialized only at the end of each trial. The generated trajectories are then input into any RL algorithm, maximizing expected reward in a trial; the paper uses TRPO, while we use A2C and PPO. As with EPOpt, our implementation of RL$^2$ is built on top of the implementations of A2C and PPO.

## 5 ENVIRONMENTS

Our environments are modified versions of four environments from the classic control problems in OpenAI Gym (Brockman et al., 2016) (CartPole, MountainCar, Acrobot, and Pendulum) and two environments from OpenAI Roboschool (Schulman et al., 2017) (HalfCheetah and Hopper) that are based on the corresponding MuJoCo (Todorov et al., 2012) environments. CartPole, MountainCar, and Acrobot have discrete action spaces, while the others have continuous action spaces. We alter the implementations to allow control of several environment parameters that affect the transition probability distribution functions of the corresponding MDPs. Each of the six environments has three versions, with $d$ parameters allowed to vary.

1. Deterministic (D): The parameters of the environment are fixed at the default values in the implementations from Gym and Roboschool. Every time the environment is reset, only the state is reset.

2. Random (R): Every time the environment is reset, the parameters are uniformly sampled from a $d$-dimensional box containing the default values. This is done by independently sampling each parameter uniformly from an interval containing the default value.

3. Extreme (E): Every time the environment is reset, its parameters are uniformly sampled from $2^d$ $d$-dimensional boxes anchored at the vertices of the box in R. This is done by independently sampling each parameter uniformly from the union of two intervals that straddle the corresponding interval in R.

Appendix A contains a schematic of the parameter ranges in D, R, and E when $d = 2$. We now describe the environments.

**CartPole** (Barto et al., 1983). A pole is attached to a cart that moves on a frictionless track. For at most 200 time steps, the agent pushes the cart either left or right with the goal of keeping the pole upright. There is a reward of 1 for each time step the pole is upright, with the episode ending when the angle of the pole is too large. Three environment parameters can be varied: (1) push force magnitude, (2) pole length, (3) pole mass.

**MountainCar** (Moore, 1990). The goal is to move a car to the top of a hill within 200 time steps. At each time step, the agent pushes a car left or right, with a reward of $-1$. Two environment parameters can be varied: (1) push force magnitude, (2) car mass.

**Acrobot** (Sutton, 1995). The acrobot is a two-link pendulum attached to a bar with an actuator at the joint between the two links. At each time step, the agent applies torque (to the left, to the right, or not at all) to the joint in order to swing the end of the second link above the bar to a height equal to the length of the link. The reward system is the same as that of MountainCar, but with a maximum of 500 time steps. We have required that the links have the same parameters, with the following three allowed to vary: (1) length, (2) mass, (3) moment of inertia.

**Pendulum**. The goal is to, for 200 time steps, apply a continuous-valued force to a pendulum in order to keep it at a vertical position. The reward at each time step is a decreasing function of the pendulum's angle from vertical, the speed of the pendulum, and the magnitude of the applied force. Two environment parameters can be varied, the pendulum's: (1) length, (2) mass.

**HalfCheetah**. The half-cheetah is a bipedal robot with eight links and six actuated joints corresponding to the thighs, shins, and feet. The goal is for the robot to learn to walk on a track without

falling over by applying continuous-valued forces to its joints. The reward at each time step is a combination of the progress made and the costs of the movements, e.g., electricity and penalties for collisions, with a maximum of 1000 time steps. Three environment parameters can be varied: (1) power, a factor by which the forces are multiplied before application, (2) torso density, (3) sliding friction of the joints.

**Hopper**. The hopper is a monopod robot with four links arranged in a chain corresponding to a torso, thigh, shin, and foot and three actuated joints. The goal, reward structure, and parameters are the same as those of HalfCheetah.

In all environments, the difficulty may depend on the values of the parameters; for example, in CartPole, a very light and long pole would be more difficult to balance. Therefore, the structure of the parameter ranges in R and E was constructed to include environments of various difficulties. The actual ranges of the parameters for each environment were chosen by hand and are listed in Appendix A.[2]

# 6 EXPERIMENTAL SETUP

In sum, we benchmark six algorithms (A2C, PPO, EPOpt-A2C, EPOpt-PPO, $RL^2$-A2C, $RL^2$-PPO) and six environments (CartPole, MountainCar, Acrobot, Pendulum, HalfCheetah, Hopper). With each pair of algorithm and environment, we consider nine training-testing scenarios: training on D, R, and E and testing on D, R, and E. We refer to each scenario using the two-letter abbreviation of the training and testing environment versions, e.g., DR for training on D and testing on R. For A2C, PPO, EPOpt-A2C, and EPOpt-PPO, we train for 15000 episodes and test on 1000 episodes. For $RL^2$-A2C and $RL^2$-PPO, we train for 7500 trials, equivalent to 15000 episodes, and test on the last episodes of 1000 trials. Note that this is a fair comparison as policies without memory of previous episodes are expected to have the same performance in any episode of a trial, and we are able to evaluate the ability of $RL^2$-A2C and $RL^2$-PPO to adapt their policy to the environment parameters of the current trial. For the sake of completeness, we do a thorough sweep of hyperparameters and randomly generate random seeds. We report results over several runs of the entire hyperparameter sweep (the only difference being the random seeds). In the following paragraphs we describe the network architectures for the policy and value functions, our hyperparameter search, and the performance metrics we use for evaluation.

**Policy and value function parameterization**. We consider two network architectures for the policy and value functions. In the first, following Henderson et al. (2018), the policy and value functions are multi-layer perceptrons (MLPs) with two hidden layers of 64 units each and hyperbolic tangent activations; there is no parameter sharing. We refer to this architecture as FF (feed-forward). In the second,[3] the policy and value functions are the outputs of two separate fully-connected layers on top of a one-hidden-layer RNN with long short-term memory (LSTM) cells of 256 units. The RNN itself is on top of a MLP with two hidden layers of 256 units each, which we call the feature network. Again, hyperbolic tangent activations are used throughout; we refer to this architecture as RC (recurrent). For A2C, PPO, EPOpt-A2C, and EPOpt-PPO, we evaluate both architectures (whose inputs are the environment states), while for $RL^2$-A2C and $RL^2$-PPO, we evaluate only the second architecture (whose input is a tuple of states, actions, rewards, and Booleans as discussed in Section 4). In all cases, for discrete action spaces policies sample actions by taking a softmax function over the policy network output layer; for continuous action spaces actions are sampled from a Gaussian distribution with mean the policy network output layer and diagonal covariance matrix whose entries are learned along with the policy and value function network parameters.

**Hyperparameters**. During training, in each algorithm and each version of each environment, we performed grid search over a set of hyperparameters used in the optimizers, and selected the value with the highest success probability when tested on the same version of the environment. The set of hyperparameters includes the learning rate for all algorithms and the length of the trajectory generated at each iteration (which we call batch size) for A2C, PPO, $RL^2$-A2C, and $RL^2$-PPO. They

---

[2]The ranges of the parameters were chosen so that a policy trained using PPO on D struggles quite a bit on the environments corresponding to the vertices of the box in R and fails completely on the environments corresponding to the most extreme vertices of the boxes in E.

[3]Based on personal communication with an author of Duan et al. (2016b).

Table 1: Generalization performance (in % success) of each algorithm, averaged over all environments (mean and standard deviation over five runs).

| Algorithm | Architecture | Default | Interpolation | Extrapolation |
|---|---|---|---|---|
| A2C | FF | $78.14 \pm 6.07$ | $76.63 \pm 1.48$ | $63.72 \pm 2.08$ |
| | RC | $81.25 \pm 3.48$ | $72.22 \pm 2.95$ | $60.76 \pm 2.80$ |
| PPO | FF | $78.22 \pm 1.53$ | $70.57 \pm 6.67$ | $48.37 \pm 3.21$ |
| | RC | $26.51 \pm 9.71$ | $41.03 \pm 6.59$ | $21.59 \pm 10.08$ |
| EPOpt-A2C | FF | $2.46 \pm 2.86$ | $7.68 \pm 0.61$ | $2.35 \pm 1.59$ |
| | RC | $9.91 \pm 1.12$ | $20.89 \pm 1.39$ | $5.42 \pm 0.24$ |
| EPOpt-PPO | FF | $85.40 \pm 8.05$ | $85.15 \pm 6.59$ | $59.26 \pm 5.81$ |
| | RC | $5.51 \pm 5.74$ | $15.40 \pm 3.86$ | $9.99 \pm 7.39$ |
| $RL^2$-A2C | RC | $45.79 \pm 6.67$ | $46.32 \pm 4.71$ | $33.54 \pm 4.64$ |
| $RL^2$-PPO | RC | $22.22 \pm 4.46$ | $29.93 \pm 8.97$ | $21.36 \pm 4.41$ |

also include the coefficient of the policy entropy in the objective for A2C, EPOpt-A2C, and $RL^2$-A2C and the coefficient of the KL divergence between the previous policy and current policy for $RL^2$-PPO. The grid values are listed in Section B. In EPOpt-A2C and EPOpt-PPO, we sample 100 environments per iteration and set $\epsilon$ first to 1.0 and then 0.1 after 100 iterations. Other hyperparameters, such as the discount factor, were set to the default values in OpenAI Baselines.

**Performance metrics**. The traditional performance metric used in the RL literature is the average total reward achieved by the policy in an episode. In the spirit of the definition of an RL agent as goal-seeking (Sutton & Barto, 2017) and to obtain a metric independent of reward shaping, we also compute the percentage of episodes in which a certain goal is successfully completed, the success rate. This additional metric is a clear and interpretable way to compare performance across conditions and environments. We define the goals of each environment as follows: CartPole: balance for at least 195 time steps, MountainCar: get to the hilltop within 110 time steps, Acrobot: swing the end of the second link to the desired height within 80 time steps, Pendulum: keep the angle of the pendulum at most $\pi/3$ radians from vertical for the last 100 time steps of a trajectory with length 200, HalfCheetah and Hopper: walk for 20 meters.

## 7 RESULTS AND DISCUSSION

We highlight some of the key findings and present a summary of the experimental results here, concentrating on the binary success metric. For each algorithm, architecture, and environment, we compute three numbers. (1) Default: success percentage on DD (the classic RL setting). (2) Interpolation: success percentages on RR. (3) Extrapolation: geometric mean of the success percentages on DR, DE, and RE. Table 1 summarizes the results. Section C contains analogous tables for each environment, which will be referred to in the following discussion.

**A2C and PPO**. With the FF architecture, the two 'vanilla' deep RL algorithms are often successful on the classic RL setting of training and testing on a fixed environment, as evidenced by the high values for Default. However, when those agents trained on environment version D are tested, we observed that they usually suffer from a significant drop in performance in R and an even further drop in E. When the algorithm is successful in the classic RL setting, as for PPO with the FF architecture, which has a Default number of 78.22, they are able to interpolate. (Interpolation equals 70.57 in that case.) That is, simply training on a distribution of environments, without any special mechanism for generalization, results in agents that can perform fairly well in similar environments. However, as expected in general they are less successful at extrapolation; PPO with the FF architecture has a Extrapolation number of 48.37. A2C with either architecture shows similar behavior to PPO with the FF architecture, while PPO with the RC architecture had difficulty training on the classic RL setting and did not generalize well. For example, on all the environments except CartPole

and Pendulum the FF architecture was necessary for PPO to train a successful policy on DD. The pattern of decrease from Default to Interpolation to Extrapolation shown in Table 1 also appears when looking at each environment individually. The magnitude of decrease depends on the combination of algorithm, architecture, and environment. For instance, on CartPole, A2C interpolates and extrapolates successfully, where Interpolation equals 100.00 and Extrapolation equals 93.63; this behavior is also shown for PPO with the FF architecture. On the other hand, on Hopper, PPO with the FF architecture has $85.54\%$ success rate in the classic RL setting but struggles to interpolate (Interpolation equals 39.68) and fails to extrapolate (Extrapolation equals 10.36). This indicates that our choice of environments and their parameter ranges led to a variety of difficulty in generalization.

**EPOpt**. With the FF architecture, EPOpt-PPO improved both interpolation and extrapolation performance over PPO, as shown in Table 1. Looking at specific environments, on Hopper EPOpt-PPO has nearly twice the interpolation performance and significantly improved extrapolation performance compared to PPO. Such an improvement also appears for Pendulum. EPOpt-PPO, similar to PPO, generally did not benefit from using the recurrent architecture; this may be due to the LSTM requiring more data to train. EPOpt however did not demonstrate the same performance gains when combined with A2C. EPOpt-A2C was able to find limited success using the RC architecture on CartPole but for other environments failed to learn a working policy even in the Default setting.

**RL$^2$**. RL$^2$-A2C and RL$^2$-PPO proved to be difficult to train and data inefficient. This is possibly due to the RC architecture, as PPO also has difficulty training on D with that architecture as shown in Table 1. On most environments, the Default numbers are low, indicating that a working policy was not found in the classic RL setting of training and testing on a fixed environment. As a result, they also have low Interpolation and Extrapolation numbers. In a few, such as RL$^2$-PPO on CartPole and RL$^2$-A2C on HalfCheetah, a working policy was found in the classic RL setting, but the algorithm struggled to interpolate or extrapolate. A success story is RL$^2$-A2C on Pendulum, where we have nearly $100\%$ success rate in DD, interpolate extremely well (Interpolation is 99.82), and extrapolate fairly well (Extrapolation is 81.79).

We observed that the partial success of these algorithms on the environments appears to be dependent on two implementation choices: the feature network in the RC architecture and the nonzero coefficient of the KL divergence between the previous policy and current policy in RL$^2$-PPO, which is intended to help stabilize training.

## 8  CONCLUSION

We introduced a new testbed and experimental protocol to measure the generalization ability of deep RL algorithms, to environments both similar to and different from those seen during training. Such a testbed enables us to compare the relative merits of algorithms for learning generalizable RL agents. Our code, based on OpenAI Gym, will be made available online and we hope that it will support future research on generalization in deep RL. Using our testbed we have evaluated two state-of-the-art deep RL algorithms, A2C and PPO, and two algorithms that explicitly tackle the problem of generalization in different ways: EPOpt, which aims to generalize by being robust to environment variations, and RL$^2$, which aims to automatically adapt to environment variations.

Overall, the 'vanilla' deep RL algorithms have better generalization performance than their more complex counterparts, being able the interpolate quite well with some extrapolation success. When combined with PPO under the FF architecture, EPOpt is able to outperform vanilla PPO; however, it does not generalize in the other cases. RL$^2$ on the other hand is difficult to train, and in its success cases provides no clear generalization advantage over the 'vanilla' deep RL algorithms or EPOpt. The sensitivity of the effectiveness of EPOpt and RL$^2$ to the base algorithm, architecture, and environment presents an avenue for future work, as intuitively EPOpt and RL$^2$ should be general-purpose approaches. We have considered model-free RL in our evaluation; another clear direction for future work is to perform a similar evaluation for model-based RL, in particular recent work such as Sæmundsson et al. (2018) and Clavera et al. (2018). Because model-based RL explicitly learns the system dynamics and generally is more data efficient, it could be better leveraged by adaptive techniques for generalization.

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

# A  ENVIRONMENT DETAILS

Table 2 details the parameter ranges for each environment and environment setting: Deterministic (D), Random (R), and Extreme (E). Figure 1 illustrates the ranges from which the parameters are sampled; the parameters for D are fixed within the range of R, and E is uniformly sampled from a range wider than R, excluding the intervals corresponding to R.

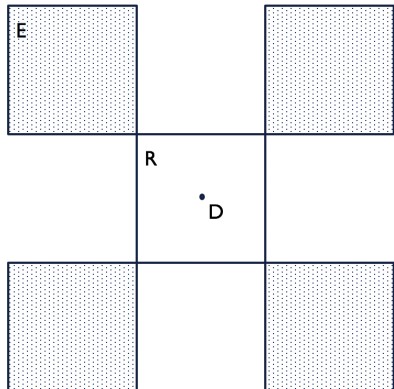

Figure 1: Schematic of the three versions of an environment.

Table 2: Ranges of parameters for each version of each environment, using set notation.

| Environment | Parameter | D | R | E |
|---|---|---|---|---|
| CartPole | Force | 10 | [5,15] | [1,5]∪[15,20] |
| | Length | 0.5 | [0.25,0.75] | [0.05,0.25]∪[0.75,1.0] |
| | Mass | 0.1 | [0.05,0.5] | [0.01,0.05]∪[0.5,1.0] |
| MountainCar | Force | 0.001 | [0.0005,0.005] | [0.0001,0.0005]∪[0.005,0.01] |
| | Mass | 0.0025 | [0.001,0.005] | [0.0005,0.001]∪[0.005,0.01] |
| Acrobot | Length | 1 | [0.75,1.25] | [0.5,0.75]∪[1.25,1.5] |
| | Mass | 1 | [0.75,1.25] | [0.5,0.75]∪[1.25,1.5] |
| | MOI | 1 | [0.75,1.25] | [0.5,0.75]∪[1.25,1.5] |
| Pendulum | Length | 1 | [0.75,1.25] | [0.5,0.75]∪[1.25,1.5] |
| | Mass | 1 | [0.75,1.25] | [0.5,0.75]∪[1.25,1.5] |
| HalfCheetah | Power | 0.90 | [0.70,1.10] | [0.50,0.70]∪[1.10,1.30] |
| | Density | 1000 | [750,1250] | [500,750]∪[1250,1500] |
| | Friction | 0.8 | [0.5,1.1] | [0.2,0.5]∪[1.1,1.4] |
| Hopper | Power | 0.75 | [0.60,0.90] | [0.40,0.60]∪[0.90,1.10] |
| | Density | 1000 | [750,1250] | [500,750]∪[1250,1500] |
| | Friction | 0.8 | [0.5,1.1] | [0.2,0.5]∪[1.1,1.4] |

# B  TRAINING HYPERPARAMETERS

The grid values we search over for each hyperparameter and each algorithm are listed below. In sum, the search space contains 183 unique hyperparameter configurations for all algorithms on a single training environment ($3,294$ training configurations), and each trained agent is evaluated on 3 test settings ($9,882$ total train/test configurations). We report results for 5 runs of the full grid search, a total of $49,410$ experiments.

- Learning rate:

- A2C, EPOpt-A2C with RC architecture, and RL$^2$-A2C: $[0.007, 0.0007, 0.00007]$
- EPOpt-A2C with FF architecture: $[0.07, 0.007, 0.0007]$
- PPO, EPOpt-PPO with RC architecture: $[0.003, 0.0003, 0.00003]$
- EPOpt-PPO with FF architecture: $[0.03, 0.003, 0.0003]$
- RL$^2$-PPO: $[0.0003, 0.00003, 0.000003]$

- Batch size:
  - A2C and RL$^2$-A2C: $[5, 10, 15]$
  - PPO and RL$^2$-PPO: $[128, 256, 512]$

- Policy entropy coefficient: $[0.01, 0.001, 0.0001, 0.00001]$

- KL divergence coefficient: $[0.3, 0.2, 0.0]$

## C  DETAILED EXPERIMENTAL RESULTS

In order to elucidate the generalization behavior of each algorithm, here we present versions of Table 1 for each environment.

Table 3: Mean and standard deviation over five runs of generalization performance (in % success) on Acrobot.

| Algorithm | Architecture | Default | Interpolation | Extrapolation |
|---|---|---|---|---|
| A2C | FF | $88.52 \pm 1.32$ | $72.88 \pm 0.74$ | $66.56 \pm 0.52$ |
|  | RC | $88.24 \pm 1.53$ | $73.46 \pm 1.11$ | $67.94 \pm 1.06$ |
| PPO | FF | $87.20 \pm 1.11$ | $72.78 \pm 0.44$ | $64.93 \pm 1.05$ |
|  | RC | $0.0 \pm 0.0$ | $0.0 \pm 0.0$ | $0.0 \pm 0.0$ |
| EPOpt-A2C | FF | $0.0 \pm 0.0$ | $0.0 \pm 0.0$ | $0.0 \pm 0.0$ |
|  | RC | $0.0 \pm 0.0$ | $0.0 \pm 0.0$ | $0.0 \pm 0.0$ |
| EPOpt-PPO | FF | $79.60 \pm 5.86$ | $69.20 \pm 1.64$ | $65.05 \pm 2.16$ |
|  | RC | $3.10 \pm 3.14$ | $6.40 \pm 3.65$ | $15.57 \pm 5.59$ |
| RL$^2$-A2C | RC | $65.70 \pm 8.68$ | $57.70 \pm 2.40$ | $57.01 \pm 2.70$ |
| RL$^2$-PPO | RC | $0.0 \pm 0.0$ | $0.0 \pm 0.0$ | $0.0 \pm 0.0$ |

Table 4: Mean and standard deviation over five runs of generalization performance (in % success) on CartPole.

| Algorithm | Architecture | Default | Interpolation | Extrapolation |
|---|---|---|---|---|
| A2C | FF | $100.00 \pm 0.0$ | $100.00 \pm 0.0$ | $93.63 \pm 9.30$ |
|  | RC | $100.00 \pm 0.0$ | $100.00 \pm 0.0$ | $83.00 \pm 11.65$ |
| PPO | FF | $100.00 \pm 0.0$ | $100.00 \pm 0.0$ | $86.20 \pm 12.60$ |
|  | RC | $65.58 \pm 27.81$ | $70.80 \pm 21.02$ | $45.00 \pm 18.06$ |
| EPOpt-A2C | FF | $14.74 \pm 17.14$ | $43.06 \pm 3.48$ | $10.48 \pm 9.41$ |
|  | RC | $57.00 \pm 4.50$ | $55.88 \pm 3.97$ | $32.53 \pm 1.47$ |
| EPOpt-PPO | FF | $99.98 \pm 0.04$ | $99.46 \pm 0.79$ | $73.58 \pm 12.19$ |
|  | RC | $29.94 \pm 31.58$ | $20.22 \pm 17.83$ | $14.55 \pm 20.09$ |
| RL$^2$-A2C | RC | $20.78 \pm 39.62$ | $0.06 \pm 0.12$ | $0.12 \pm 0.23$ |
| RL$^2$-PPO | RC | $87.20 \pm 12.95$ | $54.22 \pm 34.85$ | $51.00 \pm 14.60$ |

Table 5: Mean and standard deviation over five runs of generalization performance (in % success) on MountainCar.

| Algorithm | Architecture | Default | Interpolation | Extrapolation |
|---|---|---|---|---|
| A2C | FF | $79.78 \pm 11.38$ | $84.10 \pm 1.25$ | $89.72 \pm 0.65$ |
| | RC | $95.88 \pm 4.10$ | $74.84 \pm 6.82$ | $89.77 \pm 0.76$ |
| PPO | FF | $99.96 \pm 0.08$ | $84.12 \pm 0.84$ | $90.21 \pm 0.37$ |
| | RC | $0.0 \pm 0.0$ | $63.36 \pm 0.74$ | $15.86 \pm 31.71$ |
| EPOpt-A2C | FF | $0.0 \pm 0.0$ | $3.04 \pm 0.19$ | $3.63 \pm 0.49$ |
| | RC | $0.0 \pm 0.0$ | $62.46 \pm 0.80$ | $0.0 \pm 0.0$ |
| EPOpt-PPO | FF | $74.42 \pm 37.93$ | $84.86 \pm 1.09$ | $87.42 \pm 5.11$ |
| | RC | $0.0 \pm 0.0$ | $65.74 \pm 4.88$ | $29.82 \pm 27.30$ |
| $RL^2$-A2C | RC | $0.32 \pm 0.64$ | $57.86 \pm 2.97$ | $21.56 \pm 30.35$ |
| $RL^2$-PPO | RC | $0.0 \pm 0.0$ | $60.10 \pm 0.91$ | $31.27 \pm 26.24$ |

Table 6: Mean and standard deviation over five runs of generalization performance (in % success) on Pendulum.

| Algorithm | Architecture | Default | Interpolation | Extrapolation |
|---|---|---|---|---|
| A2C | FF | $100.00 \pm 0.0$ | $99.86 \pm 0.14$ | $90.27 \pm 3.07$ |
| | RC | $100.00 \pm 0.0$ | $99.96 \pm 0.05$ | $79.58 \pm 6.41$ |
| PPO | FF | $0.0 \pm 0.0$ | $31.80 \pm 40.11$ | $0.0 \pm 0.0$ |
| | RC | $73.28 \pm 36.80$ | $90.94 \pm 7.79$ | $61.11 \pm 31.08$ |
| EPOpt-A2C | FF | $0.0 \pm 0.0$ | $0.0 \pm 0.0$ | $0.0 \pm 0.0$ |
| | RC | $2.48 \pm 4.96$ | $7.00 \pm 10.81$ | $0.0 \pm 0.0$ |
| EPOpt-PPO | FF | $100.00 \pm 0.0$ | $77.34 \pm 38.85$ | $54.72 \pm 27.57$ |
| | RC | $0.0 \pm 0.0$ | $0.04 \pm 0.08$ | $0.0 \pm 0.0$ |
| $RL^2$-A2C | RC | $100.00 \pm 0.0$ | $99.82 \pm 0.31$ | $81.79 \pm 3.88$ |
| $RL^2$-PPO | RC | $46.14 \pm 17.67$ | $65.22 \pm 21.78$ | $45.76 \pm 8.38$ |

## D  BEHAVIOR OF MOUNTAINCAR

On MountainCar, several of the algorithms, including A2C with both architectures and PPO with the FF architecture, have greater success on Extrapolation than Interpolation, which is sometimes greater than Default (see Table 5). This is unexpected because Extrapolation combines the success rates of DR, DE, and RE, with E containing more extreme parameter settings, while Interpolation is the success rate of RR. To explain this phenomenon, we hypothesize that compared to R, E is dominated by easy parameter settings, e.g., those where the car is light but the force of the push is strong, allowing the agent to reach the top of the hill easily. In order to test this hypothesis, we create a heatmap of the reward achieved by A2C with the FF architecture trained on D and tested on R and E. We also investigated A2C with the RC architecture and PPO with the FF architecture, but because the heatmaps are qualitatively similar, we show only the heatmap for A2C with the FF architecture, in Figure 2. Referring to the structure in Figure 1, we see that the reward achieved by the policy is higher in the regions corresponding to E. Indeed, it appears that the largest regions of E are those with a large force, which enables the trained policy to push the car up the hill in less than 110 time steps, achieving the goal set in Section 6. (Note that the reward is the negative of the number of time steps taken to push the car up the hill.)

Table 7: Mean and standard deviation over five runs of generalization performance (in % success) on HalfCheetah.

| Algorithm | Architecture | Default | Interpolation | Extrapolation |
|---|---|---|---|---|
| A2C | FF | $85.06 \pm 19.68$ | $91.96 \pm 8.60$ | $40.54 \pm 8.34$ |
|  | RC | $88.06 \pm 12.26$ | $74.70 \pm 13.49$ | $42.96 \pm 7.79$ |
| PPO | FF | $96.62 \pm 3.84$ | $95.02 \pm 2.96$ | $38.51 \pm 15.13$ |
|  | RC | $20.22 \pm 17.01$ | $21.08 \pm 26.04$ | $7.55 \pm 5.04$ |
| EPOpt-A2C | FF | $0.0 \pm 0.0$ | $0.0 \pm 0.0$ | $0.0 \pm 0.0$ |
|  | RC | $0.0 \pm 0.0$ | $0.0 \pm 0.0$ | $0.0 \pm 0.0$ |
| EPOpt-PPO | FF | $99.76 \pm 0.08$ | $99.28 \pm 0.87$ | $53.41 \pm 9.41$ |
|  | RC | $0.0 \pm 0.0$ | $0.0 \pm 0.0$ | $0.0 \pm 0.0$ |
| $RL^2$-A2C | RC | $87.96 \pm 4.21$ | $62.48 \pm 29.18$ | $40.78 \pm 5.99$ |
| $RL^2$-PPO | RC | $0.0 \pm 0.0$ | $0.0 \pm 0.0$ | $0.16 \pm 0.32$ |

Table 8: Mean and standard deviation over five runs of generalization performance (in % success) on Hopper.

| Algorithm | Architecture | Default | Interpolation | Extrapolation |
|---|---|---|---|---|
| A2C | FF | $15.46 \pm 7.58$ | $11.00 \pm 7.01$ | $1.63 \pm 2.77$ |
|  | RC | $15.34 \pm 8.82$ | $10.38 \pm 15.14$ | $1.31 \pm 1.23$ |
| PPO | FF | $85.54 \pm 6.96$ | $39.68 \pm 16.69$ | $10.36 \pm 6.79$ |
|  | RC | $0.0 \pm 0.0$ | $0.0 \pm 0.0$ | $0.0 \pm 0.0$ |
| EPOpt-A2C | FF | $0.0 \pm 0.0$ | $0.0 \pm 0.0$ | $0.0 \pm 0.0$ |
|  | RC | $0.0 \pm 0.0$ | $0.0 \pm 0.0$ | $0.0 \pm 0.0$ |
| EPOpt-PPO | FF | $58.62 \pm 47.51$ | $80.78 \pm 29.18$ | $21.39 \pm 16.62$ |
|  | RC | $0.0 \pm 0.0$ | $0.0 \pm 0.0$ | $0.0 \pm 0.0$ |
| $RL^2$-A2C | RC | $0.0 \pm 0.0$ | $0.0 \pm 0.0$ | $0.0 \pm 0.0$ |
| $RL^2$-PPO | RC | $0.0 \pm 0.0$ | $0.02 \pm 0.04$ | $0.0 \pm 0.0$ |

This special case demonstrates the importance of considering a wide variety of environments when assessing the generalization performance of an algorithm; each environment may have idiosyncrasies that cause performance to be correlated with parameters. For example, Figure 3 shows a similar heatmap for A2C with the FF architecture on Pendulum, in which Interpolation is greater than Extrapolation. In this case, the policy trained on D struggles more on environments from E than on those from R, which bolsters our hypothesis.

## E  TRAINING CURVES

To investigate the effect of EPOpt and $RL^2$ and the different environment versions on training, we plotted the training curves for PPO, EPOpt-PPO, and $RL^2$-PPO on each version of each environment, averaged over the five experiment runs and showing error bands based on the standard deviation over the runs. Training curves for all algorithms and environments are available at the following link: https://drive.google.com/drive/folders/1H5aBv-Lex6WQzKI-a_LCgJUER-UQzKF4. We observe that in the majority of cases training appears to be stabilized by the increased randomness in the environments in R and E, including situations where successful policies are found. This behavior is particularly apparent for CartPole, whose training curves are shown in Figure 4 and in which all five algorithms above are able to find at least partial success. We see that especially towards the end

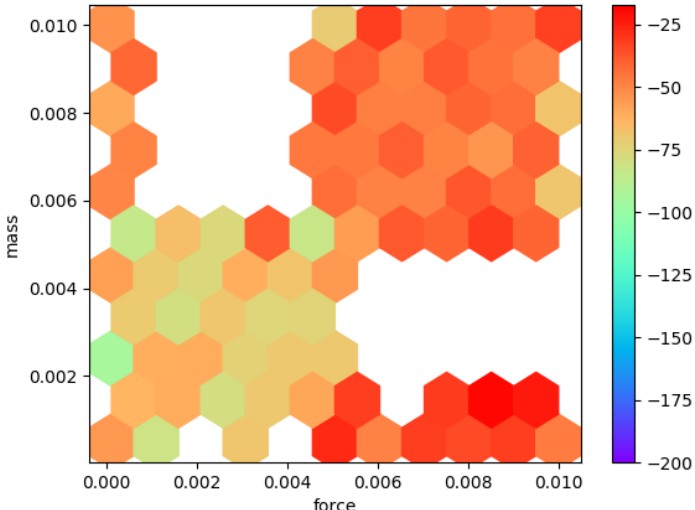

Figure 2: MountainCar: heatmap of the rewards achieved by A2C with the FF architecture on DR and DE. The axes are the two environment parameters varied in R and E.

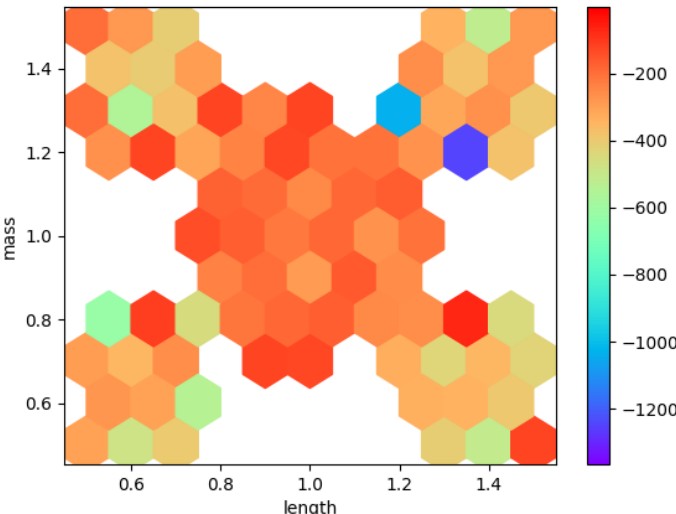

Figure 3: Pendulum: heatmap of the rewards achieved by A2C with the FF architecture on DR and DE. The axes are the two environment parameters varied in R and E.

of the training period, the error bands for training on E are narrower than those for training on D or R. Except for EPOpt-PPO with the FF architecture, the error bands for training on D appear to be the widest. Indeed, $RL^2$-PPO is very unstable when trained on D, possibly because the more expressive policy network overfits to the generated trajectories.

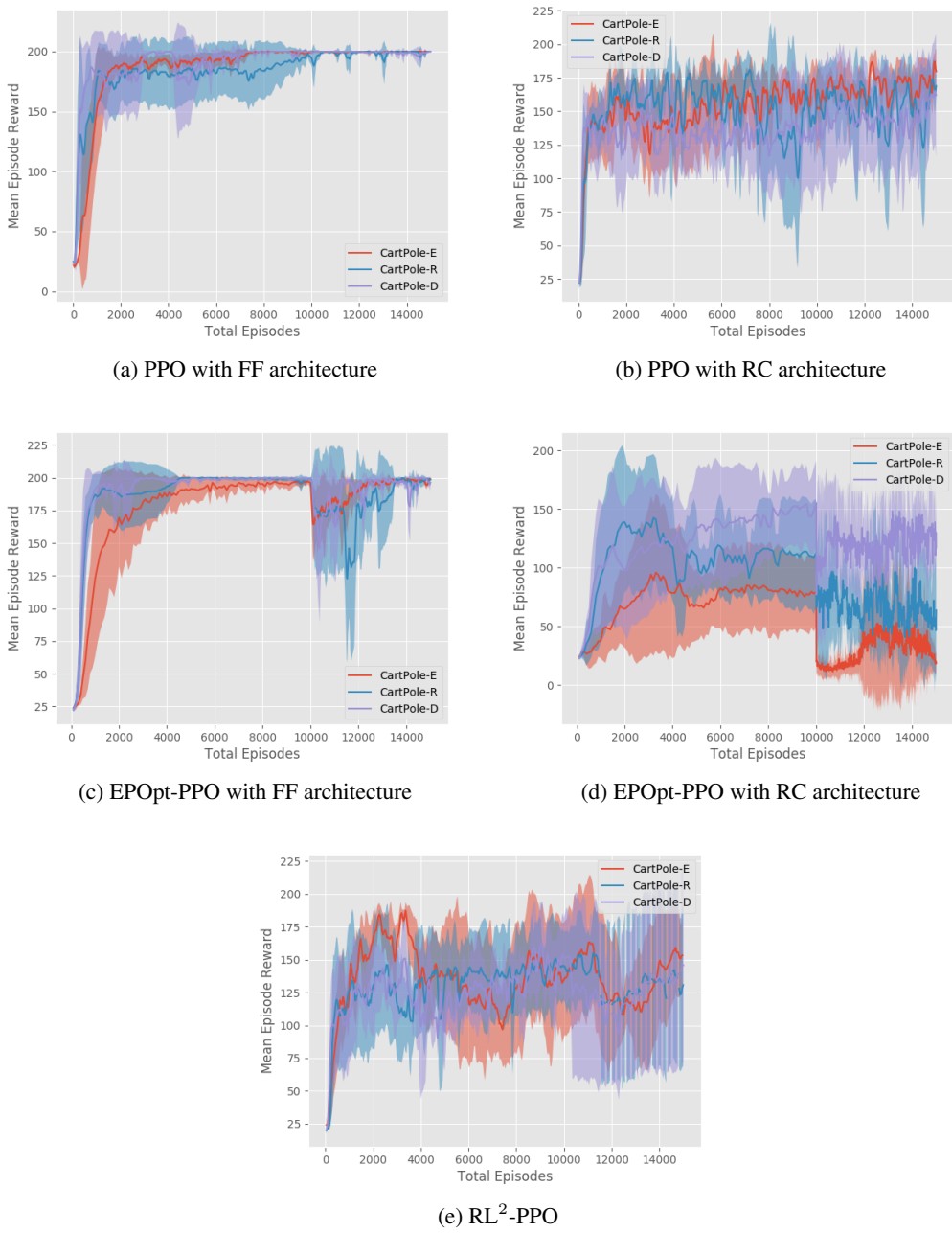

Figure 4: Training curves for the PPO-based algorithms on CartPole, all three environment versions. Note that the decrease in mean episode reward at 10000 episodes in the two EPOpt-PPO plots is due to the fact that it transitions from being computed using all generated episodes ($\epsilon = 1$) to only the 10% with lowest reward ($\epsilon = 0.1$).

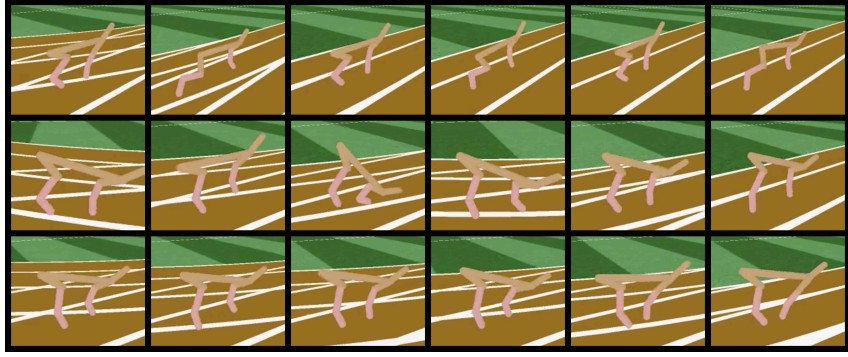

Figure 5: Video frames of agents trained with A2C on HalfCheetah, trained in the Deterministic (D), Random (R), and Extreme (E) settings (from top to bottom). All agents evaluated in the D setting.

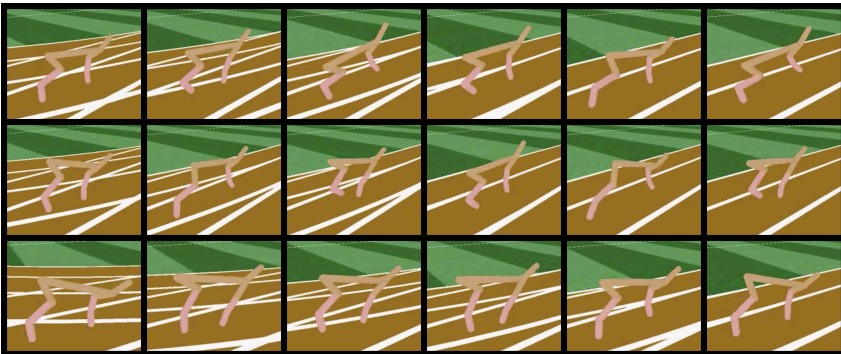

Figure 6: Video frames of agents trained with PPO on HalfCheetah, trained in the Deterministic (D), Random (R), and Extreme (E) settings (from top to bottom). All agents evaluated in the D setting.

## F  VIDEOS OF TRAINED AGENTS

The above link also contains videos of the trained agents of one run of the experiments for all environments and algorithms. We include the five scenarios considered in computing Default, Interpolation, and Extrapolation: DD, DR, DE, RR, and RE. Using HalfCheetah as a case study, we describe some particularly interesting behavior we saw.

A trend we noticed across several algorithms were similar changes in the cheetah's gait that seem to be correlated with the difficulty of the environment. The cheetah's gait became forward-leaning when trained on the Random and Extreme environments, and remained relatively flat in the agents trained on the Deterministic environment (see figures 5 and 6). We hypothesize that the forward-leaning gait developed to counteract conditions in the R and E settings. The agents with the forward-learning gait were able to recover from face planting (as seen in the second row of figure 5), as well as maintain balance after violent leaps likely caused by settings with unexpectedly high power. In addition to becoming increasingly forward-leaning, the agents' gait also tended to become stiffer in the more extreme settings, developing a much shorter, twitching stride. Though it reduces the agents' speed, a shorter, stiffer stride appears to make the agent more resistant to adverse settings that would cause an agent with a longer stride to fall. This example illustrates how training on a range of different environment configurations may encourage policies that are more robust to changes in system dynamics at test time.

