# OpenReview forum: "Assessing Generalization in Deep Reinforcement Learning"
_ICLR.cc/2019/Conference_

### Official Review · AnonReviewer3 · 2018-11-01
**Interesting topic and solid experiments**

**Rating:** 5
**Confidence:** 2

**Review:**

Update: Lower the confidence and score after reading other comments.
===

In this paper, the authors benchmark several RL algorithms on their abilities of generalization. The experiments show interpolation is somehow manageable but extrapolation is difficult to achieve.

The writing quality is rather good. The authors make it very clear on how their experiments run and how to interpret their results. The experiments are also solid. It's interesting to see that both EPOpt and RL^2, which claim to generalize better, generalize worse than the vanilla counterparts. Since the success rates are sometimes higher with more exploration, could it be possible that the hyperparameters of EPOpt and RL^2 are non-optimal?

For interpolation/extrapolation tasks, all 5 numbers (RR, EE, DR, DE, RE) are expected since the geometric mean is always 0 once any of the numbers is 0.

What does ``"KL divergence coefficient" in RL^2-PPO mean? OpenAI's Baselines' implementation includes an entropy term as in A2C.

---

> ### Author Response · Authors · 2018-11-15
> **Replying to AnonReviewer3's comments**
>
> Thank you very much for your feedback.
>
> We did a pretty thorough hyperparameter search; there are no additional hyperparameters for RL^2 compared to the PPO/A2C equivalents (apart from the added KL divergence coefficient for RL^2-PPO, and the choice of episodes-per-trial). It may be the case that RL^2 is relatively sample inefficient, however we also noticed that RL^2 is relatively volatile during training. EPOpt has two additional hyperparameters - the number of “normal” iterations before beginning to use the worst-epsilon trajectories, and epsilon. We use the corresponding values reported in the EPOpt paper (their experiments are on Hopper and HalfCheetah).
>
> In our revision we have redefined the generalization summary numbers. Our previous definition of Interpolation (geometric mean of RR and EE) and extrapolation (geometric mean of DR, DE, and RE) led to some confusing results where Interpolation=0 but Extrapolation>0 because the algorithms found it harder to train on E. We have removed EE from Interpolation and the results are now more sensible.
>
> Section 4 in the PPO paper (Schulman et al. 2017) describes the using KL divergence as a penalty in the loss function. Its coefficient is currently set to zero in the OpenAI Baselines PPO implementation, but we found that a nonzero coefficient improved training stability in RL^2-PPO, which is relatively volatile otherwise. The KL divergence coefficient becomes an additional hyperparameter in our grid search (the range does include zero which removes the penalty from the loss function).

---

### Official Review · AnonReviewer2 · 2018-11-05

**Rating:** 5
**Confidence:** 3

**Review:**

This paper proposes a benchmark for for reinforcement learning to study generalization in stationary and changing environments. A combination of several existing env. from OpenAi gym is taken and several ways to set this parameters is proposed. Paper provides a relatively thorough study of popular methodologies on this benchmark.

Overall, I am not sure there is a pressing need for this benchmark and paper does not provide an argument why there is an urgent need for one.

For instance, paragraph 3 on page 1 details a number of previous studies. Why those benchmarks are in-adequate?
On page at the end of second paragraph a  number of benchmarks from transfer learning literature is mentioned. Why not just use those and disallow model updates?
In the same way, it is not clear why new metric is introduced? How does it correlate with standard reward metrics?

Overall, as empirical study, I think this work is interesting but I think paper should justify why we need this new benchmark.

---

> ### Author Response · Authors · 2018-11-15
> **Replying to AnonReviewer2's comments**
>
> Thank you very much for your feedback.
>
> We apologize if this was not clear, but the works cited in the third paragraph of Section 1 are not benchmarks or empirical studies; they propose algorithms designed to build agents that generalize. However, they have widely varying experimental setups, both in terms of environments (e.g. MuJoCo) and their variations to which the trained agents are supposed to generalize (e.g. a heavier robot torso). They also do not use common metrics for generalization performance. Therefore, it is difficult to fairly compare them and to determine which perform best in what situations. Furthermore, many consider interpolation but not extrapolation. This was the motivation for our work.
>
> Whiteson et al. (2011) and Duan et al. (2016) cited in Section 2 are more similar to our work in that they focus on how to evaluate RL algorithms. Whiteson et al. (2011) propose a similar experimental protocol to us, differentiating interpolation and extrapolation, but consider simple tasks and tabular learning. Duan et al. (2016) appear to consider only interpolation on simple tasks (no locomotion). Neither evaluate methods specifically designed for generalization. OpenAI Retro is a benchmark for transfer learning in RL, considering Sonic the Hedgehog games, but gives information about test environment configurations during training and does not differentiate between interpolation and extrapolation.
>
> We believe that the binary success metric has several advantages. First, it makes the results much more interpretable across tasks, environment conditions, and different software implementations by separating the reported performance level from reward shaping. For example, the reward structure for HalfCheetah is different from Roboschool to RLLab and it is not at all clear how it differs for HalfCheetah-v0 and HalfCheetah-v1 in Gym. Second, it is in line with the original spirit of RL as ‘goal seeking’ discussed in Sutton and Barto (2017).

---

### Official Review · AnonReviewer4 · 2018-11-05

**Rating:** 3
**Confidence:** 5

**Review:**

This paper presents a new benchmark for studying generalization in deep RL along with a set of benchmark results. The benchmark consists of several standard RL tasks like Mountain Car along with several Mujoco continuous control tasks. Generalization is measured with respect to changes in environment parameters like force magnitude and pole length. Both interpolation and extrapolation are considered.

The problem considered in this paper is important and I agree with the authors that a good set of benchmarks for studying generalization is needed. However, a paper proposing a new benchmark should have a good argument for why the set of problems considered is interesting. Similarly, the types of generalization considered should be well motivated. This paper doesn’t do a good job of motivating these choices.

For example, why is Mountain Car a good task for studying generalization in deep RL? Mountain Car is a classic problem with a two-dimensional state space. This is hardly the kind of problem where deep RL shines or is even needed at all. Similarly, why should we care whether an agent trained on the Cart Pole task can generalize to a pole length between 2x and 10x shorter than the one it was trained on without being allowed to update its policy? Both the set of tasks and the distributions of parameters over which generalization is measured seem somewhat arbitrary.

Similarly, the restriction to methods that do not update its policy at test time also seems arbitrary since this is somewhat of a gray area. RL^2, which is one of the baselines in the paper, uses memory to adapt its policy to the current environment at test time. How different is this from an agent that updates its weights at test time? Why allow one but not the other?

In addition to these issues with the proposed benchmark, the baseline results don’t provide any new insights. The main conclusion is that extrapolation is more difficult than interpolation, which is in turn more difficult than training and testing on the same task. Beyond that, the results are very confusing. Two methods for improving generalization (EPOpt and RL^2) are evaluated and both of them seem to mostly decrease generalization performance. I find the poor performance of RL^2-A2C especially worrisome. Isn’t it essentially recurrent A2C where the reward and action are fed in as inputs? Why should the performance drop by 20-40%?

Overall, I don’t see the proposed tasks becoming a widely used benchmark for evaluating generalization in deep RL. There are just too many seemingly arbitrary choices in the design of this benchmark and the lack of interesting findings in the baseline experiments highlights these issues.

Other comments:
- “Massively Parallel Methods for Deep Reinforcement Learning” by Nair et al. introduced the human starts evaluation condition for Atari games in order to measure generalization to potentially unseen states. This should probably be discussed in related work.
- It would be good to include the exact architecture details since it’s not clear how rewards and actions are given to the RL^2 agents.

---

> ### Author Response · Authors · 2018-11-15
> **Replying to AnonReviewer4's comments**
>
> Thank you very much for your feedback.
>
> In our revision we will make it clearer that we focus on generalization to changes in the environment dynamics. Other works consider generalization in the same context, i.e., “testing performance degradation in the presence of systematic physical differences between training and test domains” (Rajeswaran et al. 2017). Whiteson et al. (2017) and Zhang et al. (2018) also consider this type of generalization. We believe that state-of-the-art algorithms should be able to solve these simpler generalization tasks (e.g., not overfitting to training domain in simulator) before addressing more complex ones such as the combinatorial generalization discussed in “Relational inductive biases, deep learning, and graph networks” by Battaglia et al. (2018).
>
> We chose these set of tasks for several reasons. They are classic tasks in RL that are implemented in the widely-used OpenAI Gym and used in previous literature on generalization in deep RL. Varying their parameters such as length and mass for Pendulum is a simple way to create environments with “systematic physical differences”. It also enables us to differentiate between interpolation and extrapolation in a way that reflects the real world; environment version R can be thought of as a distribution of normal situations and version E can be thought of as a distribution of edge cases, which are unusual in some sense. These tasks are often considered simple, but we believe that this view is because the classic RL setup considers one fixed environment configuration and that considering variations in the environment presents new challenges.
>
> The distribution of parameters for each environment version was carefully chosen by watching video footage of agents trained on environment D to determine realistic ranges for possible success (which were used to construct R), and non-realistic ranges (which were used to construct E). The binary success metric is admittedly subjective, but is also chosen carefully to correlate with what a user would consider the learning objective in a given simulator (for example, if you learned to walk, you should be able to get to 20 meters on the track). We believe that this type of metric should be supported in Gym environments because it separates policy performance evaluation from the reward shaping used for training (which may vary between different software implementations of the same environment).
>
> We agree that there is a gray area associated with our choice to assess only approaches to generalization that do not allow policy updates at test time. Our choice at the beginning of the project was motivated by a desire to do a thorough evaluation of a few methods and we decided to not include algorithms that make gradient updates to the model at test time, such as MAML.
>
> We have rewritten the introduction and conclusion to emphasize the main takeaways of our baseline evaluations. On average, the vanilla deep RL algorithms, despite their reputation for brittleness (Henderson et al. 2017), interpolate and extrapolate as well or better than EPOpt and RL^2, which are specifically designed for generalization. In other words, simply training a policy that is oblivious to environment changes on random perturbations of the default environment configuration can be very effective. The only exception is PPO with the FF architecture, where EPOpt generalizes a bit better than the vanilla algorithm.
>
> The effectiveness of EPOpt and RL^2 is highly dependent on the base algorithm (A2C or PPO) and the environment, although intuitively they should be general-purpose approaches. For instance, in most environments EPOpt appears to be effective only when combined with PPO under the FF architecture. Exploring why this occurs would be an interesting avenue for future work. We found that the training of RL^2 is less stable than that of the vanilla deep RL algorithms, possibly due to the fact that the RC policy takes as input the trajectories of multiple episodes instead of one episode; an example is shown in Figure 4. This partially explains its poorer generalization performance, but more investigation is needed to ascertain the true cause. It is important to note that while the EPOpt paper evaluates on Hopper and HalfCheetah, the RL^2 paper does not evaluate on any of the six tasks we consider.
>
> Thank you for the reference to Nair et al. (2016) We have included it in Section 2 as an early example of evaluating generalization in RL. We have expanded the description of the RC architecture to clarify the policy inputs for the two RL^2 algorithms versus the other algorithms.

---

> ### Author Response · Authors · 2018-11-29
> **Addition to our previous comment**
>
> The results of the OpenAI Retro contest are consistent with our conclusion that vanilla deep RL algorithms usually generalize better than EPOpt and RL^2. As a recap, the OpenAI Retro contest was a transfer learning challenge on Sonic the Hedgehog games. Given a set of training levels, teams were tasked with training a policy and a fast learner that could be used to fine-tune the policy given a million time steps at test time. Apart from the test-time fine-tuning, this corresponds to training on R and testing on E in our framework. The winning team’s strategy was to train a single policy using PPO on all the training levels, with each level weighted equally (i.e. vanilla PPO in our paper), and then fine-tune it using PPO at test time. A blog post with details about the contest results is found here: https://blog.openai.com/first-retro-contest-retrospective/.

---

### Author Response · Authors · 2018-11-15
**Note regarding latest revision (11/14/2018)**

Updates to results:
  - We now report results (mean and standard deviation) over five complete runs of the hyperparameter grid search.
  - We have redefined the generalization summary numbers. Our previous definition of Interpolation (geometric mean of RR and EE) and extrapolation (geometric mean of DR, DE, and RE) led to some confusing results where Interpolation=0 but Extrapolation>0 because the algorithms found it harder to train on E. We have removed EE from Interpolation (now just RR) and the results are now more sensible.
  - Section 7 (results and discussion) has been updated to match the new numbers (and the new definition of Interpolation), however the overall conclusions / themes did not change.

Updates to appendix:
  - Section D has been added which explains some unintuitive results from MountainCar.
  - Section E has been added, which investigates the effect of EPOpt and RL^2 on training. We observe that training appears to be stabilized by increased randomness in environments (R and E, vs D).
  - Section F has been added, which investigates the effect of environment difficulty on learned policies. We illustrate how training on a range of environment configurations (instead of a fixed/deterministic environment) may encourage policies more robust to changes in system dynamics at test time.
  - The complete set of training curves and evaluation videos discussed in Section E and F are available at the following (anonymous) Google drive link: https://drive.google.com/drive/folders/1H5aBv-Lex6WQzKI-a_LCgJUER-UQzKF4

---

### Meta-Review · Area_Chair1 · 2018-12-13
**Important topic, poorly motivated benchmark**

**Confidence:** 4
**Recommendation:** Reject

**Metareview:**

The manuscript proposes benchmarks for studying generalization in reinforcement learning, primarily through the alteration of the environment parameters of standard tasks such as Mountain Car and Half Cheetah. In contrast with methodological innovations where a numerical argument can often be made for the new method's performance on well-understood tasks, a paper introducing a new benchmark must be held to a high standard in terms of the usefulness of the benchmark in studying the phenomenon under consideration.

Reviewers commended the quality of writing and considered the experiments given the set of tasks to be thorough, but there were serious concerns from several reviewers regarding how well-motivated this benchmark is and restrictions viewed as artificial (no training at test-time), concerns which the updated manuscript has failed to address. I therefore recommend rejection at this stage, and urge the authors to carefully consider the desiderata for a generalization benchmark and why their current proposed set of tasks satisfies (or doesn't satisfy) those desiderata.